# Structural Characterization and Hypoglycemic Function of Polysaccharides from *Cordyceps cicadae*

**DOI:** 10.3390/molecules28020526

**Published:** 2023-01-05

**Authors:** Yani Wang, Tingting Zeng, Hang Li, Yidi Wang, Junhui Wang, Huaibo Yuan

**Affiliations:** 1College of Food and Biological Engineering, Hefei University of Technology, Hefei 230601, China; 2Sichuan Institute of Food Inspection, Chengdu 611135, China

**Keywords:** *Cordyceps cicadae*, polysaccharides, structural characteristics, HepG2 cells, insulin resistance

## Abstract

The polysaccharides isolated and purified from different parts of the medicinal fungus *Cordyceps cicadae* were identified, and three extracts displaying significant biological activities were selected for further study. The bacterium substance polysaccharides (BSP), spore powder polysaccharides (SPP), and pure powder polysaccharides (PPP) were separated, purified, and collected from the sclerotia, spores, and fruiting bodies of *Cordyceps cicadae*, respectively. The structures of *Cordyceps cicadae* polysaccharides were analyzed using gas chromatography, Fourier-transform infrared spectroscopy, methylation analysis, and one-dimensional (^1^H and ^13^C) nuclear magnetic resonance spectroscopy. Moreover, the hypoglycemic effect of *Cordyceps cicadae* polysaccharides was examined in both in vitro and in vivo models. BSP, SPP, and PPP significantly increased glucose absorption in HepG2 cells, and alleviated insulin resistance (IR) in the in vitro model. SPP was the most effective, and was therefore selected for further study of its hypoglycemic effect in vivo. SPP effectively improved body weight and glucose and lipid metabolism in type 2 diabetes model mice, in addition to exerting a protective effect on liver injury. SPP regulated the mRNA expression of key PI3K/Akt genes involved in the insulin signaling pathway. The hypoglycemic mechanism of SPP may reduce hepatic insulin resistance by activating the PI3K/Akt signaling pathway. Spore powder polysaccharides (SPP) extracted from *Cordyceps cicadae* effectively improved body weight and glucose and lipid metabolism in type 2 diabetes model mice, in addition to exerting a protective effect on liver injury. The mechanism underlying the hypoglycemic effect of SPP regulates the mRNA expression of key PI3K/Akt genes involved in the insulin signaling pathway to alleviate insulin resistance. Our results provide a theoretical basis for research into the hypoglycemic effect of *Cordyceps cicadae*, and lay the foundation for the development of functional products.

## 1. Introduction

The main manifestation of diabetes is hyperglycemia caused by a lack of insulin production or insulin resistance [1,2]. Type 2 diabetes mellitus (T2DM) is very common, making up roughly 90% of all diabetic cases [3,4,5,6]; it can lead to kidney disease, heart disease, and other complications if left unmanaged [7,8]. Drugs for the treatment of T2DM, such as DPP-4 inhibitors and sulfonylureas (SUS), have certain side effects [9,10,11]; therefore, finding effective active substances to prevent hyperglycemia is of utmost importance. Interestingly, these substances can also be derived from natural sources [12,13].

Insulin is a hormone that controls blood sugar [14]. Insulin resistance (IR) is a systemic condition defined as impaired insulin function despite elevated insulin levels (hyperinsulinemia). IR typically occurs prior to the development of T2DM [15]. Previous studies have shown that the regulation of IR in T2DM is mainly mediated by the PIK3/Akt pathway [16]; therefore, inhibiting PI3K/Akt signaling will likely reduce liver IR and increase insulin sensitivity [17]. Insulin receptor substrate 1 (IRS-1), PI3K, and glucose transporter type 4 (GLUT4) play key roles in the absorption and utilization of glucose [18,19]. The hypoglycemic mechanism is mainly related to activation of PI3K by insulin receptor substrates (IRSs) and subsequent activation of Akt [20,21], which transmits signals to downstream pathways to control cell survival and glucose and lipid metabolism [22,23]. Activation of the PI3K/Akt pathway promotes insulin-mediated glucose uptake in target tissues by increasing the transport of GLUT4 to the plasma membrane [24]. GLUT4 regulates the IRS1/2/PI3K signaling pathway, and reduces blood glucose levels via the uptake and utilization of glucose in tissues [25]. Therefore, the PI3K/Akt signaling pathway has become an important molecular mechanism for diabetes research.

*Cicada*, *Clavicipitaceae*, and *Cordyceps* are widely used as medicinal and functional foods in many Asian countries [26,27]. Cordyceps cicada can also be used to make soup or wine [28]. *Cicada Cordyceps* grows mostly in tropical areas with an appropriate temperature and a high environmental humidity. At the turn of spring and summer, *Paecilomyces cicadae* is completely parasitic on *Cicada pupae* or the larvae of *Cicada flammable*, *Platypleura kaempferi*, and *Cryptotympana pustulata*. The host larvae are consumed as nutrition and become a mass of tightly accumulated hyphae, subsequently forming a flower bud-like matrix from the mouth, head, or bottom [29].

Polysaccharides from Grifolia folsa can reduce blood glucose by inhibiting the JNK pathway [30]. Carbon nanoparticles (CPNs) infused with burdock root polysaccharides were nanoparticles that were synthesized by a hydrothermal method, and these CPNs inhibited α-glucosidase activity to reduce blood sugar in diabetic mice [31]. *Cordyceps cicadae* contains polysaccharides, adenosine, glycopeptide complexes, nucleosides, and other substances of high nutritional value [32,33]. Polysaccharides have become widely used materials due to their universality, outstanding functions, and powerful pharmacological effects [34]. *Cordyceps militaris* has many pharmacological efficacies, such as immune regulatory, anti-tumor, anti-viral, anti-infection, anti-aging, memory improving, hypoglycemic, and weight loss activities [35]. *Cordyceps cicadae* polysaccharides have strong free radical-scavenging abilities and significant antioxidant activities in vitro and as such [36], have previously been explored; however, few studies exist regarding their hypoglycemic activity. The present paper elaborates on the biological functions of *Cordyceps cicadae* polysaccharides.

## 2. Results and Discussion

### 2.1. Purified Polysaccharides

Polysaccharides were eluted using a DEAE Sepharose chromatography column to remove impurities and collect the primary polysaccharides (Figure 1A). Three purified polysaccharides, SPP, BSP, and PPP, were obtained with yields of 19.33%, 46.15%, and 34.33%, respectively.

### 2.2. Chemical Compositions and Molecular Weights of SPP, BSP, and PPP 

The chemical compositions of the polysaccharides were analyzed (Figure 1B). The total sugar contents of BSP, SPP, and PPP were 58.15%, 77.49%, and 72.62%, respectively, and their uronic acid contents were 11.35%, 21.39%, and 17.47%, respectively. In addition, BSP, SPP, and PPP contained low levels of protein. Chemical composition analysis showed that BSP, SPP, and PPP were heteropolysaccharides, and their monosaccharides were different. SPP and PPP were found to be composed of arabinose, galactose, glucose, xylose, and mannose at a molar composition ratio of 9.10:15.40:41.20:17.50:16.80 and 4.50:11.90:62.20:10.70:10.70, respectively. BSP comprised arabinose, galactose, glucose, and xylose at a molar ratio of 7.60:1.80:76.10:14.50. It was observed that the molar ratio of glucose was high, while that of arabinose was low.

### 2.3. FT-IR and UV-Vis Spectra

The UV spectra of SPP, BSP, and PPP ranged from 200 to 400 nm. No absorbance peaks were found at 280 nm or 260 nm, indicating the absence of proteins and nucleic acids. IR spectroscopy is typically used as a method to analyze the structure of polysaccharides. The three polysaccharides possessed a wide spectral band in the region of 3200–3400 cm^−1^ and strong absorption peaks in the region of 3290 cm^−1^, indicating O–H stretching vibration of polysaccharide hydroxyl groups. The alkyl C–H bond of the polysaccharide appeared as a weak absorption peak near 2930 cm^−1^. Generally, the polysaccharides possessed alkyl and hydroxyl groups, which indicates that the *Cordyceps cicadae* powders were composed of polysaccharides. The strong absorption peaks at 1643 cm^−1^ and 1409 cm^−1^ in the BSP spectrum are caused by the asymmetric and symmetric stretching vibrations of carboxyl groups or carboxylates in the polysaccharide structure, suggesting that BSP is acidic. Under the influence of pyranose, characteristic absorption peaks appeared at 1149 cm^−1^, 1078 cm^−1^, and 1016 cm^−1^. SPP and PPP had similar FT–IR absorption bands, indicating similarities in their structural features. Due to the tensile vibration of furan, an absorption peak appeared in the range of 1200–1000 cm^−1^ (Figure 1C).

### 2.4. Methylation Status

Methylation analysis is the most effective method to identify the type and proportion of polysaccharide chains (Table 1). BSP, SPP, and PPP were found to be composed of 8, 12, and 15 glycosidic residues, respectively. The composition of glycoside residues produced by BSP, SPP, and PPP is consistent with that of monosaccharides.

### 2.5. Nuclear Magnetic Resonance (NMR) Spectra

In order to study the structural characteristics of BSP, SPP, and PPP, ^1^H and ^13^C NMR spectra of *Cordyceps cicadae* polysaccharides were collected and analyzed (Figure 1D). In NMR analysis, α–Configuration related δ 95–101 ppm, β Configuration related δ 101–105 ppm [37]. The chemical shift in ^1^H−NMR was generally between 4.4−5.5 ppm, the α proton chemical shift was usually located at δ 5–6 ppm, the β proton chemical shift was located at δ 4–5 ppm [38,39,40]. In the hydrogen spectra of BSP, SPP, and PPP, some signals were found at 4–5 ppm, indicating the presence of β−configuration; a signal peak was found at 5–5.5 ppm, indicating the presence of α-configuration, and its carbon spectrum confirmed this. The signals at δ 3.2–4.0 ppm were proton peaks of the sugar skeleton, which were assigned to H2–H6 [41]. BSP, SPP, and PPP α−Glycosidic bond and β−Glycoside linkage revealed the typical characteristics of Cordyceps polysaccharides. A 4.70 ppm D2O signal can mask some signals. The ^13^C NMR spectra were also used to assign anomeric carbon regions (δ 95–110 ppm) and ring carbon regions (δ 50–85 ppm), which corresponded to the isomerization region (δ 4.5−5.5 pm) and cyclic proton region (δ 3.1–4.5 ppm) of ^1^H–NMR [42].

In the hydrogen spectra of SPP, signals in the anomeric proton corresponded to α− Galp sugar residues (5.29 ppm) [43,44]. A BSP chemical shift of 4.79 corresponded to β−Manp, SPP chemical shifts of 4.79, 5.08, and 5.04 corresponded to β−Manp, α−Araf, and α−Araf, respectively; a PPP chemical shift of 4.79 was attributed to β−Manp−(1⟶. In the ^1^H NMR spectrum, the signal of BSP in the range of 5.22–4.97 ppm and the signal peak at 4.97 ppm were attributed to anomeric protons of arabinose and glucuronic acid residues, respectively. The signal of SPP in the range of 5.18–4.95 ppm and the signal peak at 5.11–4.95 ppm were attributed to anomeric protons of arabinose and glucuronic acid residues, respectively [45]. It can be inferred from the NMR spectrum that the polysaccharide has acetal structure (90–110 ppm), methyl and methylene (3–5.5 ppm), hydroxyl (50–85 ppm), and some heteroatoms. The peaks in the range of 75–85 ppm indicate that the polysaccharide has branched chains [46].

### 2.6. In Vitro Hypoglycemic Activity

#### 2.6.1. Cytotoxicity of SPP, BSP, and PPP

The cytotoxic effects of SPP, BSP, and PPP on HepG2 cells were determined using the MTT method (Figure 2A). HepG2 cells were seeded in 96-well plates at 2 × 10^4^ cells per well and cultured at 37 °C for 24 h. Subsequently, cells were treated with 0.0125, 0.25, 0.05, 0.1, 0.2, 0.4, 0.8, 1.6, and 3.2 mg/mL of SPP, BSP, and PPP for 24 h. HepG2 cells under SPP intervention had a greater viability than those treated with BSP and PPP. When the concentrations of SPP were 0.0125, 0.25, 0.05, and 0.10 mg/mL, the survival rates of HepG2 cells were 105.43%, 100.57%, 105.00%, and 100.76%, respectively, indicating that SPP was beneficial to the growth of HepG1 cells. The survival rates of HepG2 cells were 88.10%, 88.80%, 88.14%, 90.58%, and 97.35%, and 88.10%, 88.00%, 91.47%, 95.50%, and 99% when BSP and PPP concentrations were 0.20, 0.40, 0.80, 1.60, and 3.20mg/mL, respectively. Therefore, the concentrations of the polysaccharides chosen for further experiments were 0.2, 0.4, 0.8, 1.6, and 3.2 mg/mL.

#### 2.6.2. IR-HepG2 Cell Model

At insulin concentrations of 5 × 10^−9^–5 × 10^−5^ mmol/L, there was no severe effect on cell viability (*p* > 0.05), indicating that insulin within this concentration range was not toxic to HepG2 cells; therefore, these concentrations were used in subsequent experiments (Figure 2B).

As shown in Figure 2C, in comparison with the blank group, 5 × 10^−9^, 5 × 10^−8^, and 5 × 10^−7^ mmol/ L of insulin significantly reduced glucose consumption in mice after 48 h (*p* < 0.01). After 60 h, 5 × 10^−5^ mmol/L insulin had the greatest hypoglycemic effect (*p* < 0.01). Considering that too long a modeling time would likely cause excessive cell density or changes in cell morphology, an insulin concentration of 5 × 10^−8^ mmol/L for 48 h was established to create a stable and reliable IR-HepG2 cell model.

#### 2.6.3. Effect of Cordyceps Cicadae Polysaccharides on Glucose Consumption in the IR-HepG2 Cell Model

As shown in Figure 2D, in comparison with the blank control, glucose consumption in HepG2 cells was reduced by a high concentration of insulin (*p* < 0.01), indicating good insulin resistance and successful model establishment. SPP and PPP significantly increased the glucose consumption in the cell model at concentrations of 0.2−3.2 mg/mL (*p* < 0.01), and glucose consumption was positively correlated with the concentration of glucose in the culture medium. BSP significantly increased glucose consumption at concentrations of 1.6–3.2 mg/mL (*p* < 0.01). It was observed that the three *Cordyceps cicadae* polysaccharides effectively promoted glucose metabolism in IR-HepG2 cells, and improved IR without obvious cytotoxicity. SPP had the most marked effect on increasing glucose consumption in the in vitro HepG2 cell model, followed by PPP and BSP; therefore, SPP was selected for the study of its hypoglycemic activity in vivo.

### 2.7. In Vivo Hypoglycemic Activity

#### 2.7.1. Effects of SPP on Body Weight and Blood Glucose in Mice

The body weight of mice that were fed a high-fat diet increased significantly (*p* < 0.01, *p* < 0.05). Mice in the DC group gained the most weight (32.00%). SPPL and SPPM increased body weight by 11.91% and 11.54%, respectively. Mice in the PC and SPPH groups gained little weight, roughly 6% and 8%, respectively. The effects of SPP and metformin on mice were studied in comparison with the model group. T2DM mice gained less weight following treatment with cicada extracts and metformin. Blood glucose levels in the other high-fat diet groups were significantly higher than that in the normal group (*p* < 0.01). After treatment, the blood glucose was lower than that in the model group (*p* < 0.05). The glucose level in the PC group was decreased by 46.23%, and those in the SPPL, SPPM, and SPPH groups were decreased by 5.42%, 11.78%, and 30.48%, respectively. In conclusion, SPP can improve obesity and hyperglycemia in T2DM mice (Figure 3A).

#### 2.7.2. Effect of SPP on the Organ Index in T2DM Mice

As shown in Figure 3B, the liver index in the model group treated with SPP decreased significantly (*p* < 0.01). There was no significant difference between the medium-dose, high-dose, and normal groups. The renal index in the high-dose SPP group was more significantly decreased than that in the untreated group (*p* < 0.05). In comparison with the control model group, medium and high doses of SPP significantly decreased the spleen index (*p* < 0.01); however, there was no significant difference following treatment with low-dose SPP (*p* > 0.05). After injection of metformin, the organ index in mice significantly decreased (*p* < 0.05).

#### 2.7.3. Effect of SPP on Blood Lipid Levels in T2DM Mice

The parameters of blood lipid metabolism were investigated to evaluate whether SPP can improve dyslipidemia in T2DM mice (Figure 3C). It can be seen from the blood lipid indexes in the model group that the T2DM displayed typical hyperlipidemic characteristics.

In the model group, the triglyceride level was significantly decreased, and the high-density lipoprotein cholesterol level was significantly increased in the polysaccharide treatment groups (*p* < 0.01). The medium and low total cholesterol and low-density lipoprotein cholesterol were decreased following treatment, and significantly decreased in the high-dose group (*p* < 0.01). There was no significant difference between high-density and low-density lipoprotein cholesterol in the SPPH intervention group (*p* > 0.05). Therefore, we can conclude that SPP regulates blood lipid levels in T2DM mice, and effectively improves lipid abnormalities caused by obesity, with high-dose SPP having the most marked effect.

#### 2.7.4. Effects of SPP on the Morphology of Pancreatic and Liver Tissue

Histological changes in the pancreas were studied via H&E staining (Figure 3D). In the normal control group, the structure of the pancreas was observed to be complete; the nuclei were normal and spherical in shape, there were abundant cytoplasmic granules, the structure was clear, and the arrangement was regular. In comparison, the islets in T2DM mice without treatment were damaged, showing β-cell atrophy, decreased β-cell mass, fat infiltration, structural disorder, loose tissue, and fuzzy boundaries. Observation of the metformin and SPPL groups showed a greater number of living cells, a more regular cell shape, and less fat infiltration than that seen in the SPPM and SPPH groups. The distribution of islet cells was relatively uniform, and their morphology was clear. Islet morphology in mice in the SPPH group was close to that seen in the normal group. The polysaccharides in SPP had an obvious reparation effect on the islets in T2DM mice, with high-dose SPP having the most prominent effect.

Glucolipid metabolic abnormalities can aggravate the symptoms of diabetes, and liver reactions to IR may interfere with liver glycogen transport after a meal during elevated fasting glucose, thereby hampering liver glycogen synthesis.

Liver abnormalities were studied via H&E staining. In the normal control group, hepatocytes were distributed in the linear hepatic cord, with an obvious triple-tube structure of the nucleus, central vein, and portal vein. In the T2DM group, diffuse inflammation was accompanied by severe tissue necrosis, interstitial edema, and loss of cell integrity. In the metformin treatment group, liver lobules with a clear structure and a relatively neat arrangement of hepatocytes were seen. Different doses of SPP reduced liver injury to varying degrees. In the SPPL group, cell necrosis was decreased, the structure was only slightly blurred, part of the hepatocyte cord structure recovered, and there was only a small amount of fatty degeneration in the cavity. In comparison with the SPPL group, the condition of the liver in the SPPM and SPPH groups improved, and there were no obvious pathological phenomena observed.

#### 2.7.5. Effect of SPP on the Liver Expression Levels of IRS-1, IRS-2, PI3K, Akt, and GLUT4 in T2DM Mice

The mRNA expression levels in the metformin group were the lowest in comparison with those in the other groups, while the normal group had the highest mRNA expression levels. The expression levels of these genes in liver tissue were significantly increased in the SPPM and SPPH intervention groups (Figure 3E).

#### 2.7.6. Effect of SPP on the Liver Expression of IRS-1, PI3K, Akt, and GLUT4 Proteins in T2DM Mice

The protein expression levels of IRS-1, PI3K, Akt, and GLUT4 were significantly decreased (*p* < 0.01) in the model group, at approximately 80% of that seen in normal mice. Following 28 days of treatment with SPP, the protein expression level of IRS-1 in the medium-dose group was significantly increased (*p* < 0.05). The protein expression levels of PI3K, Akt, and GLUT4 in the SPPM and SPPL groups showed a slow increasing trend. The protein expressions of IRS-1, Akt, and GLUT4 in the high-dose group increased (*p* < 0.01), reaching 90.38%, 93.23%, and 93.49% of those in the normal group, respectively. The mechanism underlying the improvement in T2DM parameters by SPP may be related to the PI3K/Akt pathway. SPP improved glucose utilization and diabetes symptoms (Figure 3F).

## 3. Materials and Methods

### 3.1. Materials and Reagents

*Cordyceps cicadae* spore powder, bacterium substance, and pure powder were all purchased from Anhui Cordyceps Biotechnology Co., Ltd. (Anhui, CHN), and were obtained using the new *Paecilomyces cicadae* strain APC-20 and patented artificial culture and fermentation technology. DEAE-Cellulose-52 was purchased from Shanghai Yuanye Biotechnology Co., Ltd. (Shanghai, CHN). Monosaccharide standard and a series of glucans of different molecular weights (180, 2700, 5250, 9750, 13,050, 36,800, 64,650, 135,350, 300,600, and 200,000 Da) were procured from Borui sugar biotechnology.

HepG2 cells were purchased from ATCC and cultured in DMEM medium containing 10% FBS and 1% penicillin (American GIBCO).

Citric acid buffer was purchased from Sinopharm Co., Ltd. (Shanghai, China)., and streptozotocin was purchased from Aladdin Reagent Company (Shanghai, China). The Rabbit IRS-1 Kit, rabbit AKT, rabbit GLUT4, and rabbit PI3K were obtained from mmbio Co., Ltd. (Shanghai, China). The analytical reagents used in the experiments were procured from Shanghai Co., Ltd. (Shanghai, China).

### 3.2. Extraction and Purification of Polysaccharides

Spore powder, bacterium substance, and pure powder (60 g) were extracted for 2 h at 78 °C in a water bath with 1.8 L distilled water and then filtered. Under the same conditions, the solid materials were extracted a further two times. All water extracts (5.4 L) were mixed, concentrated to 400 mL at 55 °C in a rotary evaporator ( Yalong re-5299, Shanghai, China), and precipitated with 80% ethanol at 4 °C. The proteins present in the polysaccharide solution were removed using Sevag reagent (chloroform: n-butanol = 5:1, *v*/*v*) until the organic layer became transparent [47]. The polysaccharide solution was dialyzed sequentially with tap water and pure water for two days. Finally, a crude polysaccharide powder was obtained by vacuum drying.

A 10 mg/mL crude polysaccharide solution of *Cordyceps cicadae* was prepared with ultrapure water, centrifuged to obtain the supernatant (3500 rpm, 10 min), and subjected to DEAE-52 column chromatography (2.6 × 40 cm). Polysaccharide fractions were eluted with different concentrations of NaCl (0, 0.1, 0.2, 0.3, 0.4, 0.5 M NaCl). Three polysaccharide fractions, SPP, BSP, and PPP, were collected, concentrated, dialyzed for 48 h, and lyophilized.

### 3.3. Chemical Composition Analysis 

Total carbohydrates in three cicada pollens were determined using the phenol sulfuric acid method [48]. The protein and total uronic acid content of the three powders was detected using the Bradford and carbazole sulfuric acid method, respectively [49,50].

### 3.4. Monosaccharide Composition Analysis

The monosaccharide compositions of SPP, BSP, and PPP were determined with ion chromatography (IC). SPP, BSP, and PPP (10 mg) were hydrolyzed with trifluoride (TFA, 3 M, 10 mL) at 120 °C for 3 h. Each solution was absorbed and transferred to a tube for drying under nitrogen. A 10 mL aliquot of water was added to vortex mix, 100 μL was absorbed, and 900 μL deionized water was added. The tube was centrifuged at 11,000 rpm for 6 min, and the supernatant was used for IC analysis. The content of the supernatant was determined on a PA20 column. Chromatography was performed using a carbon PACTM (3 mm × 150 mm, Dionex, Sunnyvale, CA, USA) IC system (Dionex ICS-5000, Thermo Fisher Scientific, Waltham, MA, USA). The retention times of standard products (Fuc, fucose; Rha, rhamnose; Ara, arabinose; Gal, galactose; Glu, glucose; Xyl, xylose; Man, mannose; Fru, fructose; Rib, ribose; GalA, galacturonic acid; GluA, glucuronic acid; GalH, galactose hydrochloride; GluH, glucosamine hydrochloride, NAD, N-acetyl-d-glucosamine; GulA, glucuronic acid; ManA, mannuronic acid) were used for monosaccharide identification.

### 3.5. Molecular Weight Analysis of SPP, BSP, and PPP

The molecular weights of SPP, BSP, and PPP were determined using a TM250 column (7.8 × 300 mm, Waters Corp, Milford, MA, USA, ) on an Agilent 1260 high-performance liquid chromatography gel with evaporative light scattering (Agilent Technology, Ltd. Palo Alto, CA, USA).

The polysaccharides were purified using deionized water at a flow rate of 1.0 mL/min. Polyethylene glycol of different molecular weights was used as a calibrator to construct standard curves and calculate the molecular weights of SPP, BSP, and PPP.

### 3.6. FT-IR and UV-Vis Spectroscopy

Mass of 1 mg of SPP, BSP, and PPP were separately mixed with 100 mg KBr powder and scanned under a Thermo Nicolet FT-IR spectrometer (Thermo Fisher Scientific Inc., Waltham, MA, USA). The wavelength range was 400–4000 cm^−1^. The three polysaccharide powders were dissolved in distilled water, and the wavelength range was set to 200−400 nm on the UV-vis spectrometer (Agilent Technologies, Palo Alto, CA, USA).

### 3.7. Methylation Analysis

The polysaccharides were methylated according to the method described by Needs et al. [51]. A 2–3 mg mass *Cordyceps cicadae* polysaccharide sample was placed in a glass reaction bottle, and 1 mL of anhydrous DMSO was added. Methylating reagent A solution was added and dissolved using ultrasound, to which methylating reagent B solution was added. The reaction was allowed to proceed via magnetic stirring in a water bath at 30 °C for 60 min. Finally, 2 mL of ultrapure water was added to terminate the methylation reaction.

### 3.8. Nuclear Magnetic Resonance (NMR) Analysis

The molecular weight of BSP is roughly 128 kDa, those of the two components of SPP are 43 kDa and 4 kDa, and those of the two components of PPP are 903 kDa, and 21 kDa. For NMR measurements, 50 mg of each of the three freeze-dried polysaccharide components (SPP, BSP, and PPP) were dissolved in 0.5 mL of deuterium oxide (D_2_O, 99.9%) [52]. Spectral ^1^H NMR and ^13^C NMR were recorded on the Bruker spectrometer (Bruker, Germany). The frequency of the ^13^C spectrum was 100.62 MHz, and the frequency of the ^1^H spectrum was 400.13 MHz. The samples were solubilized in D_2_O at a temperature of 353 K for the polysaccharide. The residual signal of the solvent was used as the internal standard: 4.25 ppm at 353 K. ^13^C spectra were recorded using 90° pulses, 20,000 Hz spectral width, 65,536 data points, 1.638 s acquisition time, 1 s relaxation delay, and between 8192 and 16,834 scans. The proton spectra were recorded with a 4006 Hz spectral width, 32,768 data points, 4.089 s acquisition times, 0.1 s relaxation delays, and 16 scans [53]. Baseline calibration was performed by Bernstein Polynomial Fit, and the polynomial order was 3.

### 3.9. In Vitro Hypoglycemic Activity Analysis

#### 3.9.1. Cytotoxicity Assays of SPP, BSP, and PPP

HepG2 cells were seeded on 96-well plates at 2 × 10^4^ cells per well, and cultured at 37 °C for 24 h. GFP-W at different concentrations (0.0125, 0.25, 0.05, 0.1, 0.2, 0.4, 0.8, 1.6, 3.2 mg/mL) was added five times. Wells without cells or polysaccharides were used as the blank and control, respectively. After incubation for 24 h, the supernatant was removed, 20 μL of MTT (5 mg/mL) solution was added, and the cells were incubated for 4 h. Subsequently, the supernatant was removed, 200 μL of DMSO was added, the purple Zan crystals were dissolved by shaking for 10 min, and the absorbance was measured at 490 nm. The cell viability was calculated as the percentage of viable cells following treatment with GFP-W in comparison with the control, according to the following equation:(1)$$\mathrm{Cell}\;\mathrm{viability}\;\left(\%\right)=\left[\left(\mathrm{OD}\;\mathrm{Sample}-\;\mathrm{OD}\;\mathrm{Blank}\right)\;/\;\left(\mathrm{OD}\;\mathrm{Control}\;-\;\mathrm{OD}\;\mathrm{Blank}\right)\right]\times 100\;\%$$

#### 3.9.2. Establishment of the IR-HepG2 Cell Model

##### Effect of Insulin Concentration on the Survival Rate of HepG2 Cells

The cells were cultured for 24 h with serum-free high-glucose DMEM containing insulin of different concentrations (10^−5^, 10^−6^, 10^−7^, 10^−8^, 10^−9^) at 37 °C and 5% CO_2_ by volume fraction. The same medium without insulin was used as the blank control. Cell viability was determined using the method described in Section 3.9.1.

##### Effect of Insulin Concentration on Glucose Consumption by HepG2 Cells

Refer to Part 1 of Section 3.9.2. for experimental methods. After culturing, the glucose content of the medium was determined using a glucose detection kit. The absorbance was measured at 505 nm, with the blank set to zero. The model with the lowest glucose consumption and the highest cell activity was used in subsequent experiments.Use the following formula to calculate glucose consumption.
Glu  (mmol/L) = A determination/A standard × five(2)

##### Effects of SPP, BSP, and PPP on Glucose Consumption by IR-HepG2 Cells

IR-HepG2 cells were divided into T2DM, metformin-treated positive control (2 mg/mL), SPP-treated (0.2, 0.4, 0.8, 1.6, 3.2 mg/mL), BSP-treated (0.2, 0.4, 0.8, 1.6, 3.2 mg/mL), and PPP-treated (0.2, 0.4, 0.8, 1.6, 3.2 mg/mL) groups. Cells were cultured in 100 μL of high-glucose DMEM medium per well. The normal control group was HepG2 cells in DMEM, and the blank group was DMEM without HepG2 cells. 

### 3.10. In Vivo Hypoglycemic Activity Analysis

#### 3.10.1. Animals and Experimental Design

Male Kunming ICR mice (74-week-old) were purchased from Jiangsu Syung Pharmaceutical Bioengineering Co., Ltd., and randomly grouped. After two weeks of acclimation, the mice were fasted for 14 h. The experimental group was injected with 100 mg/kg STZ to establish the T2DM model. Fasting blood glucose was measured one week later. The T2DM model was considered successfully established at a blood glucose level ≥ 16.7 mmol/L.

T2DM mice were divided into five groups: model, positive control (PC; 200 mg/kg bw^−1^/day metformin), spore powder polysaccharide low-dose (SPPL; 100 mg/kg), spore powder polysaccharide medium-dose (SPPM; 200 mg/kg), and spore powder polysaccharide high-dose (SPPH; 400 mg/kg) groups. 

#### 3.10.2. Fasting Blood Glucose (FBG) and Serum Lipid Level Analysis

Blood glucose was measured using an electronic glucose meter (GB type, Roche Diabetes Care GMBH, Mannheim, GER) after modeling, and 2 and 4 weeks post-intragastric administration. Serum lipid levels were measured using ELISA.

#### 3.10.3. Histopathological Analysis and Organ Index Calculation

The liver, kidney, and pancreas were subjected to hematoxylin and eosin (H&E) staining, and examined under a microscope (Nikon Eclipse E100) equipped with a digital camera (Nikon ds-u3). The organs were weighed, and the organ index was calculated using the following equation:Organ index (%) = organ mass/mouse body weight × 100%(3)

#### 3.10.4. Quantitative Real-Time Polymerase Chain Reaction (RT-qPCR)

IRS-1, IRS-2, PI3K, Akt, and GLUT4 were amplified using RT-qPCR. The primers used were as follows: IRS-1 For, 5′-TGGCAGGAGAGTGGTGGAGTTG-3′; IRS-1 Rev, 5′-GGTAGGAGGTGTCGGAGAAGAAGA-3′; IRS-2 For, 5′-GGAGCAACACACCCGAGTCAATAG-3′; IRS-2 Rev, 5′-GAAGAGACCATCAAGTCCAGCGGA-3′; Akt For, 5′-AGAGGCAGGAAGAAGAGACGATGG-3′; Akt Rev, 5′-CGAATGACTCTTGGCACAGGACG-3′; PI3K For, 5′-AGAAAGGTGTGCGGCAGAAGAAG-3′; PI3K Rev, 5′-CGATACGGACGAGGCATCACCAT-3′; GLUT4 For, 5′-GCTGGTGTGGTCAATACGGTCTTC-3′; GLUT4 Rev, 5′-GATAAACGGCAGGAGGACGAACC-3′; β-actin For, 5′-CACGATGGAGGGGcCGGACTCATC-3′; β-actin Rev, and 5′-TGACACAACCGTATCTCCAGAAAT-3′. The lengths of amplified IRS-1, IRS-2, Akt, PI3K, GLUT4, and β-actin were 270 bp, 283 bp, 276 bp, 233 bp, 392 bp, and 300 bp, respectively. Samples were homogenized using a mortar and pestle with liquid nitrogen, and total RNA was isolated from the liver using TRIzol™ reagent (Shanghai Shenggong Bioengineering Co., Ltd, Shanghai, China). The Fsq-301 cDNA Reverse Transcription Kit (Transgen Biotechnology Co., Ltd., Beijing, China) was used for cDNA synthesis, and Taq PCR Master Mix (Sangen Biotechnology Co., Ltd., Shanghai, China) was used for RT-qPCR. mRNA expression was normalized to that of β-actin.

#### 3.10.5. Determination of Protein Expression Levels

Levels of IRS-1, PI3K, Akt, and GLUT-4 in the liver tissue of each group were determined using specific ELISA kits ( Quanshijin Biotechnology Co., Ltd., Beijing, China).

### 3.11. Statistical Analysis

The experimental results are expressed as the mean ± standard deviation (SD). Statistical significance was analyzed using one-way analysis of variance (ANOVA) using SPSS 21.0 software. Duncan’s Multiple Range test was used for multiple comparisons of the mean. A *p* value < 0.05 was considered statistically significant. All samples were evaluated in triplicate.

## 4. Conclusions

Three polysaccharides were isolated and purified from *Cordyceps cicadae* spore powder, bacterium substance powder, and pure powder, and named SPP, BSP, and PPP, respectively. The yields of purified polysaccharides from the bacterium substance power, spore powder, and pure powder of *Cordyceps cicadae* were 34.33%, 19.33%, and 46.15%, respectively. The molecular weight of BSP was roughly 128 kDa, those of the two components of SPP were 43 kDa and 4 kDa, and those of the two components of PPP were 903 kDa and 21 kDa. The total sugar contents of BSP, SPP, and PPP were 58.15%, 77.49%, and 72.62%, respectively, and the uronic acid contents were 11.35%, 21.39%, and 17.47%, respectively. These three kinds of polysaccharides all contained low levels of protein and nucleic acids, which is consistent with the UV spectra results. SPP contained 0.19% of protein. The chemical composition of the three polysaccharide powders consisted of ara, gal, glu, and xyl, with SPP and PPP containing one mannose. All three polysaccharides contained pyranose, and existed in α and β configurations. Regarding the in vivo hypoglycemic activity, the three polysaccharides significantly increased the absorption of glucose by HepG2 cells, and alleviated IR. Amongst them, SPP displayed the most marked effect. Regarding their in vivo hypoglycemic activity, medium and high doses of SPP significantly upregulated the mRNA expression levels of *IRS-1*, *IRS-2*, *PI3K*, and *GLUT4* in the PI3K/Akt insulin signaling pathway, while high doses of SPP significantly increased the protein expression levels of IRS-1, PI3K, and GLUT4. The hypoglycemic mechanism of SPP may activate the upstream and downstream signaling molecules of the PI3K/Akt pathway to reduce liver insulin resistance.

## Figures and Tables

**Figure 1 molecules-28-00526-f001:**
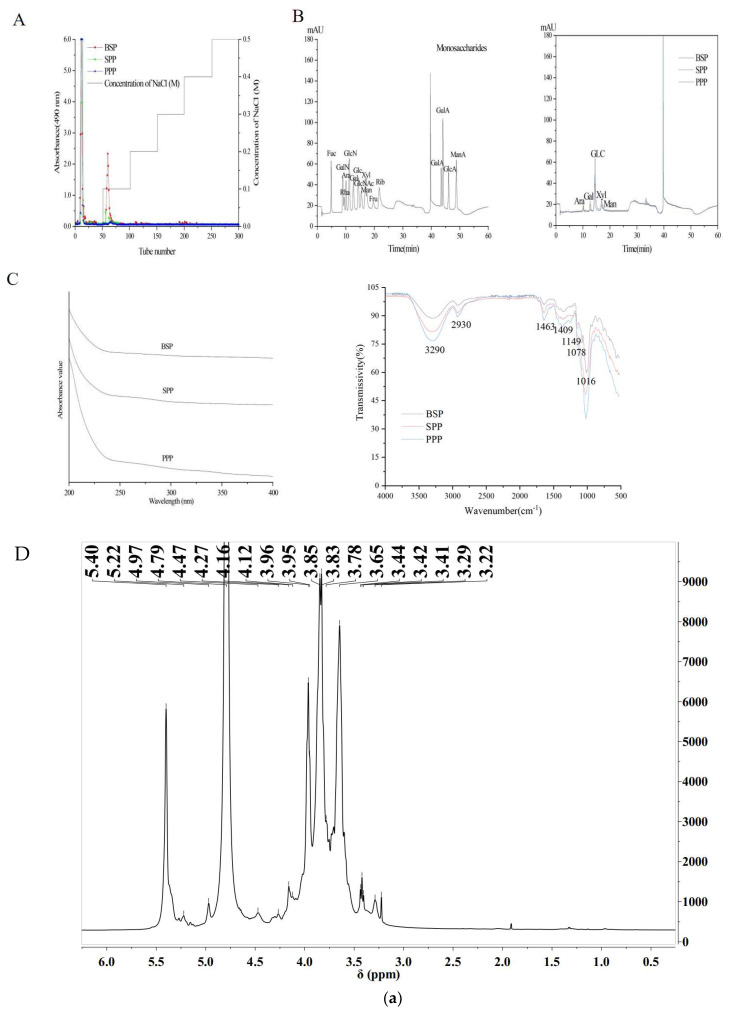
Structural characterizations of polysaccharides. (**A**) Elution curves of crude polysaccharide fractions from SPP, BSP, and PPP on a DEAE Cellulose–52 column. (**B**) Ion chromatograms of sixteen standard monosaccharides and monosaccharide compositions of SPP, BSP, and PPP. (**C**) FT–IR spectra of SPP, BSP and PPP. (**D**) Nuclear Magnetic Resonance (NMR) Spectra. ^1^H NMR (**a**–**c**) and ^13^C NMR (**d**–**f**) spectra of BSP, SPP, and PPP.

**Figure 2 molecules-28-00526-f002:**
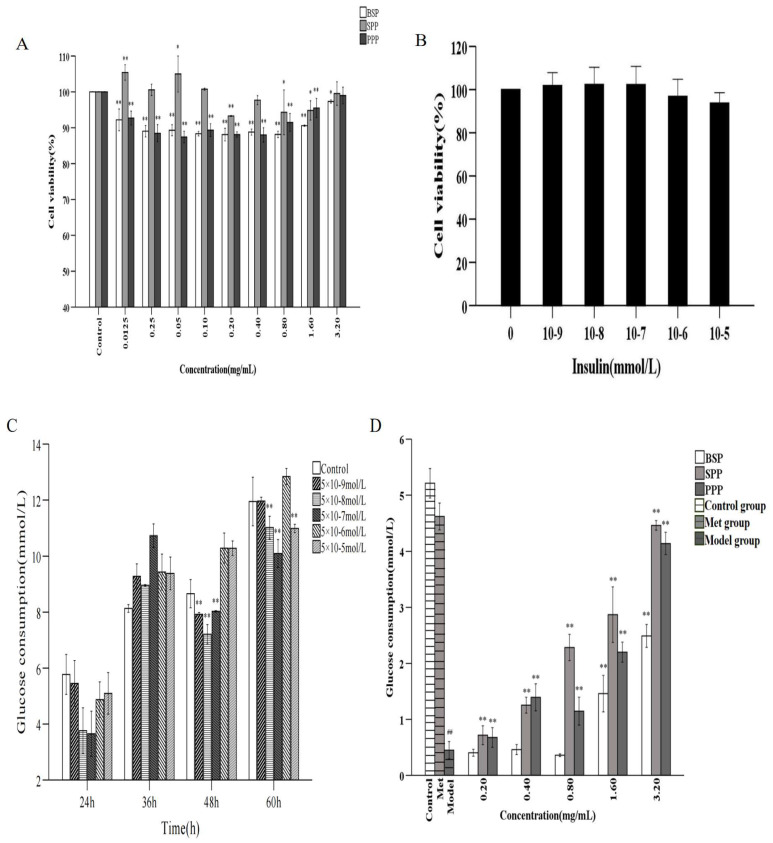
Analysis of hypoglycemic activity in vivo. (**A**) Effects of SPP, BSP, and PPP on HepG2 cytotoxicity. Note: ** (*p* < 0.01) and * (*p* < 0.05) indicate the difference between the experimental group and the blank group. (**B**) Effects of different insulin concentrations on HepG2 cells. (**C**) Effects of different insulin concentrations on IR HepG2 cell model. Note: ** (*p* < 0.01) indicate the difference between the experimental group and the blank group. (**D**) Effects of polysaccharides on glucose consumption of IR HepG2 cells. Control: control group; Model: insulin resistance model group; Met: positive control group. Note: ** (*p* < 0.01) show that there was a significant difference between the polysaccharide group and the model group; ## (*p* < 0.01) represent significant differences between the model group and the blank group.

**Figure 3 molecules-28-00526-f003:**
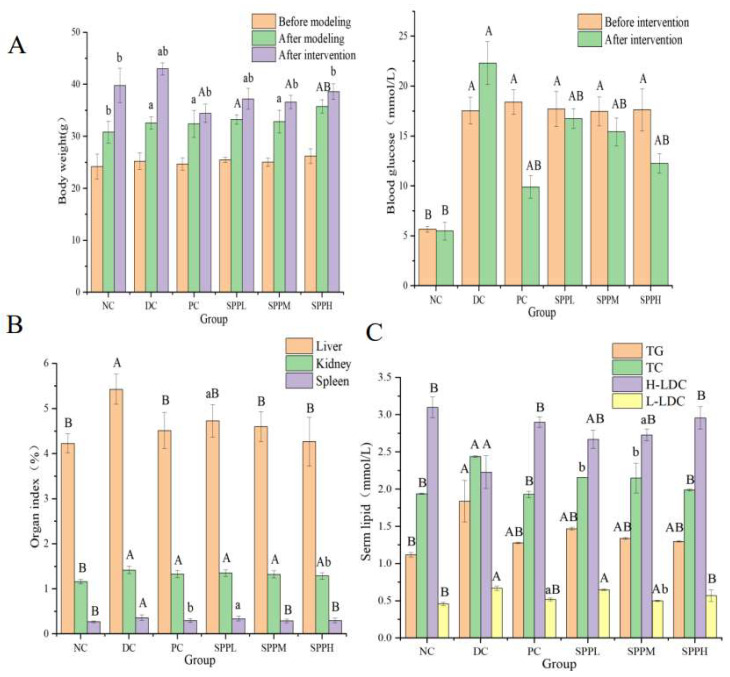
In vitro hypoglycemic activity analysis. (**A**) Effects of SPP on body weight and blood glucose of type 2 diabetic mice. (**B**) Effect of SPP on organ index of type 2 diabetic mice (**C**) Serm lipid levels in each group of mice. (**D**) Histocyte staining in mice. (**D1**) Hematoxylin-eosin staining of liver tissue, (**D2**) Hematoxylin-eosin staining of pancreatic tissue. A: NC group; B: DC group; C: PC group; D: SPPL group; E: SPPM group; F: SPPH group. (**E**) Effects of SPP on the expressions of IRS-1, IRS-2, PI3K, Akt, and GLUT4 genes in diabetes mice. (**F**) Protein expressions of IRS-1, PI3K, Akt, and GLUT4. Note: a (*p* < 0.05) and A (*p* < 0.01) represent the ratio of normal group, while b (*p* < 0.05) and B (*p* < 0.01) represent the ratio of negative control group (**A**–**C**,**F**).

**Table 1 molecules-28-00526-t001:** Methylation analysis of SPP, BSP, and PPP.

Name	RT	Methylated Sugar	The Molar Rat(%)	Type of Linkage
SPP	17.567	2,3,5-Me3-Araf	0.194	Araf-(1⟶
	19.151	2,3,4-Me3-Xylp	0.005	Xylp-(1⟶
	23.73	2,3-Me2-Araf	0.372	⟶5)-Araf-(1⟶
	25.77	2,3,4,6-Me4-Manp	0.029	Manp-(1⟶
	26.899	2,3,4,6-Me4-Galp	0.006	Galp-(1⟶
	28.309	2-Me1-Araf	0.095	⟶3,5)-Araf-(1⟶
	30.974	3,4,6-Me3-Manp	0.097	⟶2)-Manp-(1⟶
	31.849	2,3,6-Me3-Glcp	0.107	⟶4)-Glcp-(1⟶
	33.141	2,3,4-Me3-Manp	0.011	⟶6)-Manp-(1⟶
	38.611	2,3-Me2-Glcp	0.009	⟶4,6)-Glcp-(1⟶
	39.054	2,3-Me2-Galp	0.018	⟶4,6)-Galp-(1⟶
	39.441	3,4-Me2-Manp	0.056	⟶2,6)-Manp-(1⟶
BSP	17.556	2,3,5-Me3-Araf	0.058	Araf-(1⟶
	23.676	2,3-Me2-Araf	0.050	⟶5)-Araf-(1⟶
	25.729	2,3,4,6-Me4-Glcp	0.060	Glcp-(1⟶
	28.276	2-Me1-Araf	0.017	⟶3,5)-Araf-(1⟶
	31.164	2,4,6-Me3-Glcp	0.017	⟶3)-Glcp-(1⟶
	31.837	2,3,6-Me3-Glcp	0.737	⟶4)-Glcp-(1⟶
	35.936	2,6-Me2-Glcp	0.014	⟶3,4)-Glcp-(1⟶
	38.985	2,3-Me2-Glcp	0.046	⟶4,6)-Glcp-(1⟶
PPP	17.632	2,3,5-Me3-Araf	0.112	Araf-(1⟶
	19.213	2,3,4-Me3-Xylp	0.007	Xylp-(1⟶
	23.802	2,3-Me2-Araf	0.159	⟶5)-Araf-(1⟶
	25.852	2,3,4,6-Me4-Manp	0.049	Manp-(1⟶
	26.974	2,3,4,6-Me4-Galp	0.006	Galp-(1⟶
	28.393	2-Me1-Araf	0.036	⟶3,5)-Araf-(1⟶
	31.09	3,4,6-Me3-Manp	0.140	⟶2)-Manp-(1⟶
	31.296	2,4,6-Me3-Glcp	0.005	⟶3)-Glcp-(1⟶
	31.46	2,3,6-Me3-Galp	0.014	⟶4)-Galp-(1⟶
	32.086	2,3,6-Me3-Glcp	0.258	⟶4)-Glcp-(1⟶
	32.48	2,3,4-Me3-Glcp	0.006	⟶6-Glcp-(1⟶
	33.23	2,3,4-Me3-Manp	0.021	⟶6)-Manp-(1⟶
	35.968	2,6-Me2-Glcp	0.023	⟶3,4)-Glcp-(1⟶
	39.172	2,3-Me2-Glcp	0.043	⟶4,6)-Glcp-(1⟶
	39.594	3,4-Me2-Manp	0.123	⟶2,6)-Manp-(1⟶

## Data Availability

The research data used to support the findings of this study are included in the article.

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
