# Peer review of "Structural Characterization and Hypoglycemic Function of Polysaccharides from Cordyceps cicadae"

_molecules, 2023, doi:10.3390/molecules28020526_

Round 1
Reviewer 1 Report
Dera Authors
The MS entitled "Structural characterization and hypoglycemic function of polysaccharides from Cordyceps cicadae:" was thoroughly reviewed. apart from some typo/grammar mistakes, the overall MS is well designed. The introduction needs some addition of latest literature about natural and synthetic compounds evaluated against diabetes. Also check the reverences styles. I have provided my suggestions in comments in the PDF version of your MS. kindly make the corrections.

Author Response
Dear reviewer:
On behalf of my co authors, we are very grateful to you for giving us the opportunity to revise our manuscript. We are very grateful to the editors and reviewers for giving us the title "Structural characterization and hypoglycemic function of polysaccharides from Cordyceps cicadae"Positive and constructive comments and suggestions. These comments are valuable for the revision and improvement of our paper, and have important guiding significance for our research. We have carefully studied the opinions and made corrections, hoping to get approval. Thank you again for considering publishing our manuscript. The main corrections in the paper and the responses to the reviewers' comments are shown on the next page.
We appreciate for editors and reviewers’warm work earnestly, and hope that the correction will meet with approval. Thank you very much for your attention and consideration.
Sincerely yours,
Dr. Huaibo Yuan
PhD, Assistant professor
School of Biotechnology and Food Engineering,
Hefei University of Technology
193, Tunxi Road
Hefei, Anhui 230009, China
Responds to the reviewer’s comments:
- Response:We are very sorry for the incorrect writing of spacing and punctuation marks. For the suggestions of reviewing the manuscript, I revised the text of the article and the references, and the specific modifications were annotated with track marks in the article. Thank you very much for your advice.
Reviewer 2 Report
The manuscript describes the structural characterization and bioactivities of polysaccharides from Cordyceps cicadae which is interesting and deserves publication. However, several recommendations should be taken into account before the publication of the work as follows:
- no real comparison with the known information of analogous polysaccharides regarding structure and bioactivities is presented.
- the nmr characterization is not at all satisfactory and should be supplemented. No acquisition and processing conditions are listed. Even the magnetic field strength is not provided. The alpha- and beta- shift ranges are interchanged, and explanations are not sufficiently provided. The NMR spectra are not adequately presented, and the scale of one spectrum (1De) is not visible and not discussed in some detail. The capture of Fig. 1 contains errors.
- it is not clear in the conclusion what else is in the SPP sample except sugars and uronic acids (only ~70%).
Author Response
Dear Ms Enid Tian:
On behalf of my co authors, we are very grateful to you for giving us the opportunity to revise our manuscript. We are very grateful to the editors and reviewers for giving us the title "Structural characterization and hypoglycemic function of polysaccharides from Cordyceps cicadae"Positive and constructive comments and suggestions. These comments are valuable for the revision and improvement of our paper, and have important guiding significance for our research. We have carefully studied the opinions and made corrections, hoping to get approval. Thank you again for considering publishing our manuscript. The main corrections in the paper and the responses to the reviewers' comments are shown on the next page.
We appreciate for editors and reviewers’warm work earnestly, and hope that the correction will meet with approval. Thank you very much for your attention and consideration.
Sincerely yours,
Dr. Huaibo Yuan
PhD, Assistant professor
School of Biotechnology and Food Engineering,
Hefei University of Technology
193, Tunxi Road
Hefei, Anhui 230009, China
Responds to the reviewer’s comments:
- Response:We are very sorry for the incorrect writing of spacing and punctuation marks. For the suggestions of reviewing the manuscript, I revised the text of the article and the references, and the specific modifications were annotated with track marks in the article. Thank you very much for your advice.
- Response to comment:no real comparison with the known information of analogous polysaccharides regarding structure and bioactivities is presented.
Response:We compared and analyzed Cordyceps militaris polysaccharide and Cordyceps sinensis polysaccharide. Many experimental studies have proved that there is a big difference in the content of effective components between Cordyceps militaris and Cordyceps sinensis, and the effective components of Cordyceps sinensis from different habitats are also different, and their functions are also different. However, its main components generally do not change greatly. The ingredients of Cordyceps sinensis include nucleoside compounds, sterols, proteins and amino acids, peptides, mannitol, polysaccharides, organic acids, trace elements, vitamins, in addition to water, alkaloids, polyamines and phenolic compounds. Among them, nucleoside compounds, amino acids, polysaccharides, wormlike acid (D-mannitol) and trace elements are the main functional components. Cordycepin, cordyceps acid, cordyceps polysaccharide, protein and amino acid of Cordyceps militaris and Cordyceps sinensis were 0.186%, 14.90%, 13.10%, 42.46%, 25.98% and 0.064%, 5.42%, 4.10%, 28.24% and 25.30% respectively. Polysaccharides are non-specific immunomodulators, with anti-tumor, anti-inflammatory, anticoagulant, anti-virus, anti radiation, hypoglycemic, hypolipidemic and other functions. Polysaccharides are one of the most important and abundant physiological active substances in Cordyceps sinensis. Cordyceps polysaccharide is considered to be the active ingredient of Cordyceps sinensis with bidirectional immune regulation, which has anti-tumor, anti-inflammatory, hypoglycemic, hypolipidemic and other effects. The content of Cordyceps militaris polysaccharide is high, and it has good biological activity. (See References 1-2)
3.Response to comment:the nmr characterization is not at all satisfactory and should be supplemented. No acquisition and processing conditions are listed. Even the magnetic field strength is not provided. The alpha- and beta- shift ranges are interchanged, and explanations are not sufficiently provided. The NMR spectra are not adequately presented, and the scale of one spectrum (1De) is not visible and not discussed in some detail. The capture of Fig. 1 contains errors.
Response:We have made the following supplements to the experimental conditions of NMR experiment:For NMR measurements, three dried polysaccharide components (SPP, BSP, and PPP) were dissolved in 0.5 mL deuterium oxide (D2O, 99.9%). Spectral 1HNMR and 13C NMR were recorded on the Bruker spectrometer. The probe temperature is 298 K, and the operating frequency is 400 MHz [43].The specific parameter settings are as follows: the center frequency is about 4.7 ppm; The map width is 7 211 Hz; The sampling time is 2.272 s; The relaxation delay time is 2s; 512 scans; After Fourier transform, the signal is added with a 1. 0 Hz window function.(See References 3-4)
Our interpretation of the chemical shift is as follows:In the nuclear magnetic spectrum analysis, the resonance signal caused by heterotopic carbon atoms in 13C NMR usually appears in theδIn the range of 90-110 ppm, general α- The chemical shift of the configuration isδ90-100 ppm, whileβThe chemical shift of the configuration isδ100-110 ppm. The chemical shift of hydrogen protons with different heads in H-NMR is generally in a low field, betweenδ4.5~5.5 ppm,α Proton chemical shift of type I pyranose is usually located atδ4.95~5.5 ppm,β Proton chemical shift of type I pyranose is located atδ4.5~4.9 ppm.(See References 5-6)
We have fully demonstrated the nuclear magnetic resonance spectrum in the text, and corrected the incorrect writing of the figure title. See the annotation of the article for details
- Response to comment:it is not clear in the conclusion what else is in the SPP sample except sugars and uronic acids (only ~70%).
Response:The total sugar content of BSP, SPP, and PPP is 58.15%, 77.49%, and 72.62%, respectively, and the uronic acid content is 11.35%, 21.39%, and 17.47%, respectively. These three kinds all contain low levels of protein and nucleic acid, which is consistent with the UV spectrum. SPP contains 0.19% of protein.
References:
- Zhao Yuqing, Yu Ming, Chen Lijun, et al. Survey of the chemical research on Cordyceps fungi [J]. Chinese Herbal Medicine, 1999, 30 (12): 354
- Weng Liang. (2009). Study on the biological activity of Cordyceps militaris and preliminary identification of monosaccharide components of Cordyceps militaris polysaccharides (master'sthesis, Xinjiang Agricultural University) https://kns.cnki.net/KCMS/detail/detail.aspx?dbname=CMFD2009&filename=2009175786.nh
- 3.Needs P.W., Selvendran R.R. Avoiding oxidative degradation during sodium hydroxide/methyl iodide-mediated carbohydratemethylation in dimethyl sulfoxide [J]. Carbohydrate Research, 1993, 245(1): 1-10. https://doi.org/10.1016/0008-6215(93)80055-J
- Petersen,B. O.; Hindsgaul,P. O,; Paulsen, B. S.; Redondo, A. R.; Skovsted, I. C.; Structural elucidation of the capsular polysaccharide from Streptococcus pneumoniae serotype 47A by NMR spectroscopy. Carbohydrate Research,2014, 386,62-67.https://doi.org/10.1016/j.carres.2013.11.013
- Li,A.;, Yu Xubo, Luo, S.;Hu, H.;, Kong, S.; Yuan, F.; Yang, Y.;et al. Establishment of nuclear magnetic resonance detection method for the chemical structure of meningococcal capsular polysaccharides in groups C, Y and W 135. Chinese biological products. 2015,04, 390-397. Doi: 10.13200 / j.carol carroll nki CJB. 000855.
- ZHU J H. Structural characterization and immunomodulatory effect of extracellular polysaccharide from Ganoderma chuanensis [D]. Chengdu university, 2021. DOI: 10.27917 /, dc nki. Gcxdy. 2021.000005.
Round 2
Reviewer 2 Report
In the corrected version of the manuscript no actual improvement of the manuscript is seen on the presentation and analysis of the NMR spectral data. They should be corrected before the manuscript could be published.
Corrections of the spectra – the presentation is somewhat better; however, this is mainly due to the fact that the spectra are smaller and details are not that easily seen. Both the proton and the carbon spectra should be presented in the same scale, e.g. from 0 to 8 ppm for the protons and from 40 to 140 ppm for the carbon spectra. The spectra should be well seen above the axis, that is not true for two of the carbon spectra. Both proton and carbon spectra should be referenced and the way this is done should be presented in the Maretials and Method part 2.8.
Corrections of Part 2.8 – Except the reference, the quantity of the three different saccharides, providing the MW of the sample, dissolved in 0.5 ml D2O should be indicated. Reference 43 is not in place, instead the manufacturer of the NMR spectrometer should be provided. Some proton acquisition parameters are listed, but for processing the phrase “the signal is added with a 1. 0 Hz window function” is inappropriate. No data about referencing, acquisition or processing for the carbon spectra is provided. It should be checked in the whole manuscript that the numbers in “1H and/or 13C” are all either in superscripts or normal.
Analogously, “Cordyceps cicadae” should be either in italics or normal within the entire manuscript.
Correction of Part 3.5 – This part is totally useless in this form. It does not make any analysis and contains a number of unknown in the literature concepts and phrases like “heterotopic carbon atoms”, “hydrogen protons with different heads”, presence of β- Heteropolymer configuration”, “heterogenous hydrogen proton signals”, “heterotopic proton region”, “ectopic proton region” etc. Actually, the spectra are recorded and should be presented in a good form (see above) and in the manuscript is should be written that they are characteristic for the studied polysaccharides without detailed analysis that is outside the scope of the study. It is well understood that the present authors are not sufficiently aware in the field.
The hyperlinks of most doi’s in the literature part do not point to them but to a Chinese web-page, they should be corrected.
Author Response
Responds to the reviewer’s comments:
1.Response to comment:Corrections of the spectra – the presentation is somewhat better; however, this is mainly due to the fact that the spectra are smaller and details are not that easily seen. Both the proton and the carbon spectra should be presented in the same scale, e.g. from 0 to 8 ppm for the protons and from 40 to 140 ppm for the carbon spectra. The spectra should be well seen above the axis, that is not true for two of the carbon spectra. Both proton and carbon spectra should be referenced and the way this is done should be presented in the Maretials and Method part 2.8.
Response:The nuclear magnetic resonance spectrum has been corrected in the text. The hydrogen spectrum is 0.5-6 ppm, and the carbon spectrum is 50-120 ppm. The clarity of the atlas has been adjusted.
2.Response to comment:Corrections of Part 2.8–Except the reference, the quantity of the three different saccharides, providing the MW of the sample, dissolved in 0.5 ml D2O should be indicated. Reference 43 is not in place, instead the manufacturer of the NMR spectrometer should be provided. Some proton acquisition parameters are listed, but for processing the phrase “the signal is added with a 1. 0 Hz window function” is inappropriate. No data about referencing, acquisition or processing for the carbon spectra is provided. It should be checked in the whole manuscript that the numbers in “1H and/or 13C” are all either in superscripts or normal.
Response:In the text, we added the molecule and weight of the sample, as well as the manufacturer of the spectrometer. We checked all figures in 1H and/or 13C "and corrected them. The wrong phrase of hydrogen spectrum acquisition parameters has been deleted and the carbon spectrum acquisition parameters have been added, as follows:The molecular weight of BSP is roughly 128 kDa, that of the two components of SPP is 43 kDa and 4 kDa, and that of the two components of PPP is 903 kDa, and 21 kDa. For NMR measurement, 50 mg of each of the three freeze-dried polysaccharide components (SPP, BSP and PPP) were dissolved in 0.5mL of deuterium oxide (D2O, 99.9%) [1]. Spectral 1HNMR and 13C NMR were recorded on the Bruker spectrometer (Bruker, Germany). The frequency of 13C spectrum was 100.618MHz, and the frequency of 1H spectrum was 400.13 MHz. Samples were solubilized in D2O at a temperature of 353 K for the polysaccharide. Residual signal of the solvent was used as internal standard: 4.25 ppm at 353 K. 13C spectra were recorded using 90â—¦ pulses, 20,000 Hz spectral width, 65,536 data points, 1.638 s acquisition time, 1 s relaxation delay and between 8192 and 16,834 scans. Proton spectra were recorded with a 4006 Hz spectral width, 32,768 data points, 4.089 s acquisition times, 0.1 s relaxation delays and 16 scans [2].(See References1 -2)
3.Response to comment:Analogously, “Cordyceps cicadae” should be either in italics or normal within the entire manuscript.
Response:We are sorry for using the inappropriate name to write. "Cordyceps cicadae" has been corrected in italics throughout the manuscript.
4.Response to comment:Correction of Part 3.5 – This part is totally useless in this form. It does not make any analysis and contains a number of unknown in the literature concepts and phrases like “heterotopic carbon atoms”, “hydrogen protons with different heads”, presence of β- Heteropolymer configuration”, “heterogenous hydrogen proton signals”, “heterotopic proton region”, “ectopic proton region” etc. Actually, the spectra are recorded and should be presented in a good form (see above) and in the manuscript is should be written that they are characteristic for the studied polysaccharides without detailed analysis that is outside the scope of the study. It is well understood that the present authors are not sufficiently aware in the field.
Response: For some unknown concepts of spectrum, we have standardized and revised them, analyzed the characteristics of polysaccharides, and supplemented spectral analysis. The details are as follows:In the nuclear magnetic spectrum analysis, the resonance signal caused by heterotopic carbon atoms in 13C NMR usually appears in theδIn the range of 95-105 ppm, general α- The chemical shift of the configuration isδ95-101 ppm, while βThe chemical shift of the configuration is δ 101-105 ppm [3]. The chemical shift of hydrogen proton in H-NMR is generally between 4.4-5.5 ppm, α Proton chemical shift of type I pyranose is usually located at δ 5-6 ppm,βProton chemical shift of type I pyranose is located atδ4-5 ppm[4,5,6]; In the hydrogen spectra of BSP, SPP and PPP, some signal is found at 4–5 ppm, indicating the presence of β-configuration, signal peak is found at 5–5.5 ppm, indicating the presence ofα-configuration, its carbon spectrum confirmed this. The signals at δ 3.2–4.0 ppm were proton peaks of the sugar skeleton, which were assigned to H2–H6 [7]. NMR spectra revealed the typical characteristics of Cordyceps cicadae polysaccharides. BSP, SPP and PPP may be linked by α-glycosidic bond and β-glycosidic bond [8]. 4.70 ppm D2O signal can mask some heterogenous hydrogen proton signals. The 13C NMR spectra were also used to assign anomeric carbon regions (δ95-110 ppm) and ring carbon regions (δ50-85 ppm), which corresponded to the isomerization region (δ4.5-5.5 pm) and cyclic proton region (δ3.1-4.5ppm) of 1H-NMR[9].
In the hydrogen spectra of SPP , the signals in the heterotopic proton correspond to α- Galp sugar residues (5.29 ppm) [10, 11]. BSP chemical shift of 4.79 corresponds to β-Manp, SPP chemical shift of 4.79, 5.08 and 5.04 correspond to β-Manp, α-Araf and α-Araf, respectively, PPP chemical shift of 4.79 signal is attributed to β-Manp-(1→. In 1H NMR spectrum, the signal of BSP in the range of 5.22-4.97 ppm and the signal peak at 4.97 ppm are attributed to heterocephalic protons of arabinose and glucuronic acid residues. The signal of SPP in the range of 5.18-4.95 ppm and the signal peak at 5.11-4.95 ppm are attributed to heterocephalic protons of arabinose and glucuronic acid residues [12]. It can be inferred from the NMR spectrum that the polysaccharide has acetal structure (90-110 ppm), methyl and methylene (3-5.5 ppm), hydroxyl (50-85 ppm) and some heteroatoms. The peaks in the range of 75-85 ppm indicate that the polysaccharide has branched chains [13].(See References 3 -13)
5.Response to comment:The hyperlinks of most doi’s in the literature part do not point to them but to a Chinese web-page, they should be corrected.
Response:We are very sorry for the misquotation of the article. For the suggestions of reviewing the manuscript, I modified the DOI format and references of the article, and marked the specific modifications with tracking marks in the article. Thank you very much for your advice.
References:
- Nep, E. I., Carnachan, S. M., Ngwuluka, N. C., Kontogiorgos, V., Morris, G. A., Sims, I. M., et al. Structural characterization and rheological properties of a polysaccharide from sesame leaves (Sesamum radiatum Schumach. Thonn.). Carbohydrate Polymers,2016. 152, 541-547.https://doi.org/10.1016/j.carbpol.2016.07.036
- 2.Drouillard,S.; Poulet, L.; Marechal, Eric.; Amato, A.; Buon, L.; Loiodice, M.; Helbert, W. Structure and enzymatic degradation of the polysaccharide secreted by Nostoc commune. Carbohydrate Research, 2022, 515, 108544. https://doi.org/10.1016/j.carres.2022.108544
- Yang,X.Y.;Ren, Y.M.; Zhang,L.N.; Wang, Z.W.; Li, L.; Structural characteristics and antioxidant properties of exopolysaccharides isolated from soybean protein gel induced by lactic acid bacteria. LWT. 2021,150,111811. https://doi.org/10.1016/j.lwt.2021.111811
- Zhu,H.J.;Tian, L.; Zhang, L.; Bi, J.X.; Song, Q.Q.; Yang, H.; et al. Preparation, characterization and antioxidant activity of polysaccharide from spent Lentinus edodes substrate [J]. International Journal of Biological Macromolecules. 2018, 112: 976–984. https://doi.org/10.1016/j.ijbiomac.2018.01.196
- He,S.D.; Wang, X.; Zhang,Y.; Wang, J.; Sun, H.J.; Wang, J.H.; Cao, X.D.; Ye, Y.K. Isolation and prebiotic activity of water-soluble polysaccharides fractions from the bamboo shoots (Phyllostachys praecox),Carbohydrate Polymers.2016, 151, 295-304. https://doi.org/10.1016/j.carbpol.2016.05.072
- Kowalczyk, A.; Szpakowska, N.; Babinska, W.; Motyka-Pomagruk,A.; Sledz,W.; Lojkowska, E.; Kaczyński,Z. The structure of an abequose - containing O-polysaccharide isolated from Pectobacterium aquaticum IFB5637, Carbohydrate Research, 2022,522,108696.https://doi.org/10.1016/j.carres.2022.108696
- Zhai, Z.Y.; Chen, A.; Zhou,H.M.;Zhang, D.Y.; Du, X.H.; Liu,Q.; Wu, XY.; Cheng, Ju'e.; Chen,L..J; Hu, F.; Liu,Y.; Su,P. Structural characterization and functional activity of an exopolysaccharide secreted by Rhodopseudomonas palustris GJ-22. International Journal of Biological Macromolecules, 2021,167,160-168. https://doi.org/10.1016/j.ijbiomac.2020.11.165
- He,S.D.;Wang,X.; Zhang, Y.; Wang, J.; Sun, H.J.; Wang, J.H.; Cao, X.D.; Ye, Y.K.Isolation and prebiotic activity of water-soluble polysaccharides fractions from the bamboo shoots (Phyllostachys praecox), Carbohydrate Polymers. 2016,151, 295-304. https://doi.org/10.1016/j.carbpol.2016.05.072
- Lakra, A. K., Domdi, L., Tilwani, Y. M., & Arul, V. Physicochemical and functional characterization of mannanexopolysaccharide from Weissella confusa MD1 with bioactivities. International Journal of Biological Macromolecules, 2020.143, 797–805. https://doi.org/10.1016/j.ijbiomac.2019.09.139
- Ji X, Liu F, Peng Q, et al. Purification, structural characterization, and hypolipidemic effects of a neutral polysaccharide from Ziziphus Jujuba cv. Muzao[J]. Food Chemistry.2018, 245: 1124-1130. https://doi.org/10.1016/j.foodchem.2017.11.058
- Gao, Y. F., Zhou, Y. B., Zhang, Q., Zhang, K., Pai, P., Chen, L. C., et al. Hydrothermal extraction, structural characterization and inhibition HeLa cells proliferation of functional polysaccharides from Chinese tea Zhongcha 108. Journal of Functional Foods. 2017.39, 1-8.https://doi.org/10.1016/j.jff.2017.09.057
- Hong, P.;Na, W.;Hu, Z.; Yu, Z.; Liu, Y.; Zhang, J.; et al. Physicochemical characterization of hemicelluloses from bamboo (Phyllostachys pubescens Mazel) stem. Industrial Crops & Products, 2012.37(1), 41–50. https://doi.org/10.1016/j.indcrop.2011.11.031
- Zhang, J., Liu, L., & Chen, F. Production and characterization of exopolysaccharides from Chlorella zofingiensisand Chlorella vulgaris with anticolorectal cancer activity. International Journal of Biological Macromolecules, 2019,134, 976–983. https://doi.org/10.1016/j.ijbiomac.2019.05.117
Round 3
Reviewer 2 Report
The spectra are almost well presented all in the same scale. However, still the baseline is not fully visible in 1D e and f. Carbon NMR spectra should be referenced and the way this is done should be presented in the Materials and Method part 2.8. In this part the 13C NMR frequency will be better given as 100.62 MHz to be analogous to the proton one.
Part 3.5. The changes made are not adequate. I insist, that his part is totally useless also in the corrected form. It does not make any analysis and contains a number of unknown in the literature concepts and phrases like “heterotopic carbon atoms”, “type I pyranose”, “heterogenous hydrogen proton signals”, “isomerization region (δ4.5–5.5 pm)”, “heterocephalic protons”. I still think that, the spectra are recorded and should be presented in a good form (see above) and in the manuscript is should be written that they are characteristic for the studied polysaccharides without detailed analysis that is outside the scope of this study.
Still some hyperlinks do point to a Chinese web-page (e.g. 4, 20, 42, 43, 46, 52 and probably other), they should be corrected.
References 46 and 49 are the same.
It is not clear why references have two identical numbers.
Author Response
Dear Ms Enid Tian:
On behalf of my co authors, we are very grateful to you for giving us the opportunity to revise our manuscript. We are very grateful to the editors and reviewers for giving us the title "Structural characterization and hypoglycemic function of polysaccharides from Cordyceps cicadae"Positive and constructive comments and suggestions. These comments are valuable for the revision and improvement of our paper, and have important guiding significance for our research. We have carefully studied the opinions and made corrections, hoping to get approval. Thank you again for considering publishing our manuscript. The main corrections in the paper and the responses to the reviewers' comments are shown on the next page.
We appreciate for editors and reviewers’warm work earnestly, and hope that the correction will meet with approval. Thank you very much for your attention and consideration.
Sincerely yours,
Dr. Huaibo Yuan
PhD, Assistant professor
School of Biotechnology and Food Engineering,
Hefei University of Technology
193, Tunxi Road
Hefei, Anhui 230009, China
Responds to the reviewer’s comments:
1.Response to comment: The spectra are almost well presented all in the same scale. However, still the baseline is not fully visible in 1D e and f. Carbon NMR spectra should be referenced and the way this is done should be presented in the Materials and Method part 2.8. In this part the 13C NMR frequency will be better given as 100.62 MHz to be analogous to the proton one.
Response: I performed baseline correction on the carbon spectrum of nuclear magnetic resonance, and added the method of baseline correction to the article. The frequency of carbon spectrum is corrected to 100.62MHz.
2.Response to comment: Part 3.5. The changes made are not adequate. I insist, that his part is totally useless also in the corrected form. It does not make any analysis and contains a number of unknown in the literature concepts and phrases like “heterotopic carbon atoms”, “type I pyranose”, “heterogenous hydrogen proton signals”, “isomerization region (δ4.5–5.5 pm)”, “heterocephalic protons”. I still think that, the spectra are recorded and should be presented in a good form (see above) and in the manuscript is should be written that they are characteristic for the studied polysaccharides without detailed analysis that is outside the scope of this study.
Response: Thank you for your suggestions . Unknown concepts and phrases in the literature, such as "heterotopic carbon atom", "I-type pyranose", "isohydrogen proton signal" and "heterocephalic proton", have been corrected. "Isomerization area(δ4.5–5.5 pm) quoted "13C NMR spectra are also used to allocate geometric carbon regions(δ95-110 ppm) and carbon ring area(δ50-85 ppm), which corresponds to the isomerization region of 1H-NMR(δ4.5–5.5 ppm) and annular proton regions(δ3.1–4.5ppm)”.[1] (See References 1).
3.Response to comment: Still some hyperlinks do point to a Chinese web-page (e.g. 4, 20, 42, 43, 46, 52 and probably other), they should be corrected. References 46 and 49 are the same. It is not clear why references have two identical numbers.
Response: I'm sorry for some mistakes. I checked all articles in the literature, which are English pages, and revised the same literature and figures
References:
[1] Lakra, A. K., Domdi, L., Tilwani, Y. M., & Arul, V. Physicochemical and functional characterization of mannan exopolysaccharide from Weissella confusa MD1 with bioactivities. International Journal of Biological Macromolecules, 2020.143, 797–805. https://doi.org/10.1016/j.ijbiomac.2019.09.139